# Point-Cloud Completion with Pretrained Text-to-image Diffusion Models

**Yoni Kasten**[1]          **Ohad Rahamim**[2]          **Gal Chechik**[1,2]

[1]NVIDIA Research          [2]Bar-Ilan University

## Abstract

Point-cloud data collected in real-world applications are often incomplete, because objects are being observed from specific viewpoints, which only capture one perspective. Data can also be incomplete due to occlusion and low-resolution sampling. Existing approaches to completion rely on training models with datasets of predefined objects to guide the completion of point clouds. Unfortunately, these approaches fail to generalize when tested on objects or real-world setups that are poorly represented in their training set. Here, we leverage recent advances in text-guided 3D shape generation, showing how to use image priors for generating 3D objects. We describe an approach called SDS-Complete that uses a pre-trained text-to-image diffusion model and leverages the text semantics of a given incomplete point cloud of an object, to obtain a complete surface representation. SDS-Complete can complete a variety of objects using test-time optimization without expensive collection of 3D data. We evaluate SDS-Complete on a collection of incomplete scanned objects, captured by real-world depth sensors and LiDAR scanners. We find that it effectively reconstructs objects that are absent from common datasets, reducing Chamfer loss by about 50% on average compared with current methods. Project page: *https://sds-complete.github.io/*

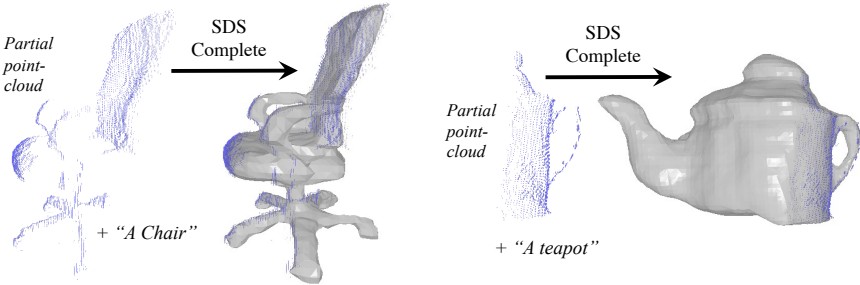

Figure 1: We present SDS-Complete: A test-time optimization method for completing point clouds captured by depth sensors, leveraging pre-trained text-to-image diffusion model. The inputs to our method are an incomplete point cloud (blue) along with a textual description of the object. The output is a complete surface (gray) that is consistent with the input points (blue). The method works well on a variety of objects captured by real-world point-cloud sensors.

## 1   Introduction

Modeling 3D objects and scenes is becoming a central part of machine perception. Most 3D data is collected using sensors that capture the 3D structure of various objects, like LIDAR or depth scanners.

37th Conference on Neural Information Processing Systems (NeurIPS 2023).

When used in the real world, various factors may cause scanners to capture incomplete or partially sampled 3D objects. First and foremost, objects are often captured from specific camera viewpoints, collecting points from only "one side" of an object (Figure 1). Reconstruction may also suffer from self-occlusions and low sensor resolution. To fully understand the three-dimensional world, one must deal with partial data and missing parts.

Current approaches for completing partial point clouds [58, 50, 30, 38, 53, 56, 49] operate as follows. They first gather extensive training datasets of complete 3D objects (e.g., [9]); then, they extract partial point clouds from the 3D objects as training data; finally, they train deep models to predict completed objects using their known ground-truth geometry. These methods are commonly evaluated on test partitions derived from their training data and consistently show high accuracy. However, they tend to generalize poorly to real data, due to several reasons. First, the training data primarily consists of Computer-Aided Designed (CAD) models, which differ from real-world objects [1]. Second, points are sampled in an artificial way, which does not accurately simulate real-world capture processes. Finally, the training data is mostly aligned [51], which can pose challenges when dealing with real-world data that is typically not perfectly aligned with the training data. The performance of these models deteriorates even further for objects and shapes that were not observed during training (label shift). This is a severe problem because the diversity of shapes in existing datasets of 3D objects is very limited. Recent work [56] attempted to address this limitation by expanding the range of object categories. However, the quality of completed objects is significantly decreased when applied to real-world data, as we illustrate below. The poor generalization of these methods to real data limits their practical use in real-world situations that demand 3D perception, such as indoor-scene reconstruction and autonomous driving.

Here, we address the challenge of completing 3D objects in the wild from real-world partial point clouds. This is achieved by leveraging priors about object shapes that are encoded in pretrained text-to-image diffusion models. Our key idea is that since text-to-image diffusion models were trained on a vast number of diverse objects, they contain a strong prior about the shape and texture of objects, and that prior can be used for completing object missing parts. For example, given a partial point cloud, knowing that it corresponds to a chair can guide the completion process, because objects from this class are expected to exhibit some types of symmetries and parts that are captured in 2D images.

A similar intuition has been used for generating 3D objects "from scratch" (DreamFusion) [39]. DreamFusion uses the SDS loss, which measures agreement between 2D model prior and renderings of the 3D shape. Unfortunately, naively applying the SDS loss to our problem of point cloud completion fails. This is because, as we show below, it does not combine well the hard constraints implied by the points collected from the sensor with the prior embedded in the diffusion model.

To address these challenges, we introduce SDS-Complete: a method to complete a given partial point cloud using several considerations. First, we use a Signed Distance Function (SDF) surface representation [37, 18, 5, 55], and constrain the zero level set of the SDF to go through the input points. Second, we use information about areas with no collected points, to rule out object parts in these areas. Third, we use a prior about camera position and orientation and a curriculum of out-painting when sampling camera positions. Finally, we use the SDS loss [39] to incorporate prior guided by the class of an object on the rendered images.

We demonstrate that SDS-Complete generates completions for various objects with different shape types from two real-world datasets: the Redwood dataset [12], which contains various incomplete real-world depth camera partial scans, and the KITTI dataset [8], which contains object LiDAR scans from driving scenarios. In both cases, SDS-Complete outperforms the current state-of-the-art methods.

In summary, this paper makes the following contributions: (1) A formulation of point cloud completion as a test-time optimization problem, avoiding the cost of collecting large datasets of 3D geometries and training models. (2) A new approach to PC completion, which combines an empirical point-cloud, with image priors using an SDF surface representation. (3) A practical and unified approach to completing and preserving real-world captured 3D content from various depth sensors, (LiDAR or depth camera) all while incorporating prior knowledge of camera poses through a well-structured camera curriculum. (4) We demonstrate state-of-the-art completion results for diverse in-the-wild objects, captured by real-world sensors.

## 2 Related work

**Surface Completion from Point Clouds.** Over the last years approaches based on deep-networks [58, 50, 30, 38] have demonstrated remarkable capabilities in reconstructing objects from incomplete or partial inputs. Early attempts with neural networks [13, 14, 19, 46] used voxel grid representations of 3D geometries due to their straightforward processing with off-the-shelf 3D convolutional layers. While voxels proved to be useful, they suffer from a space complexity issue, as their representation grows cubically. Consequently, these methods can only generate shapes with limited resolution. In contrast, point cloud representations [15, 2] have been leveraged to model higher-resolution geometries using neural networks. Several methods [52, 57] use such techniques for predicting the completed point cloud given a partial input point cloud. However, to obtain a surface representation, a surface reconstruction technique [24] needs to be applied as a post-processing step, which can introduce additional errors. Recently, an alternative approach has emerged where the output surface is represented using neural representations [30, 37]. The advantage of these representations lies in their ability to represent the surface continuously without any discretization. [30, 38] trained deep neural networks and latent conditioned implicit neural representations on a dataset of predefined object classes [9], to perform point cloud completion.

While most deep methods for surface completion train a different model per object class, very recent methods have focused on training multi-class models, allowing for better generalization [53, 56]. PoinTr [56] uses a transformer encoder-decoder architecture for translating a given input point cloud into a set of point proxies. These point proxies are then converted into completed point clouds using FoldingNet [54]. ShapeFormer [53] directly reconstructs surfaces from incomplete point clouds using a transformer. Recently, [3] combined point cloud and image inputs for completing the object shape. They further use the input image for applying weakly-supervised loss on the rendered output.

Other recent works [11, 27, 34] show progress in the task of shape completion given a partial surface, where [34] uses a transformer and autoregressive modeling, and [11, 27] employ diffusion processes that allow controlling the completion with text. However, these methods require a surface as input and cannot handle incomplete point clouds. Furthermore, their applicability is limited to the domain they are trained on.

In contrast to the above-mentioned methods, our method performs point cloud completion as a test-time optimization process using pre-trained available diffusion models, does not rely on any collection of 3D shapes for training, and works on much broader domains.

**3D models from text using 2D supervision.** Several approaches used large vision-and-language models like CLIP [40] to analyze and synthesize 3D objects. Text2Mesh [32], CLIP-Mesh [25] and DreamFields [23] present approaches for editing meshes, generating 3D models, and synthesizing NeRFs [33] respectively, based on input text prompts. The methods employ differentiable renderers to generate images while maximizing their similarity with the input text in CLIP space.

More directly relevant to the current paper are methods that build on diffusion models. Text-guided image diffusion models [43, 44, 6, 16] generate images based on text prompts, enabling control over the generated visual content. These 2D models can then be used to guide 3D object generation, an approach first presented by DreamFusion [39] and then further improved [28]. Latent-NeRF [31] enables DreamFusion to run with higher-resolution images by optimizing NeRF with diffusion model features instead of RGB colors. TEXTure [42] and Text2Tex [10] use depth-aware text-to-image diffusion models to synthesize textures for meshes. Other recent works predict shapes directly from 2D images [41, 48, 29]. In contrast, our method uses the input text for completing partial point clouds, rather than editing or synthesizing 3D content.

## 3 Method

### 3.1 Problem Setup

We address the problem of completing a surface given incomplete point cloud measurements, captured by a real point cloud sensor. In contrast to previous works for object completion [56, 53] that first trained one feed-forward model on a large dataset of 3D shapes, we operate in a test-time-optimization, and solve each object separately from scratch without using any 3D dataset for pre-training.

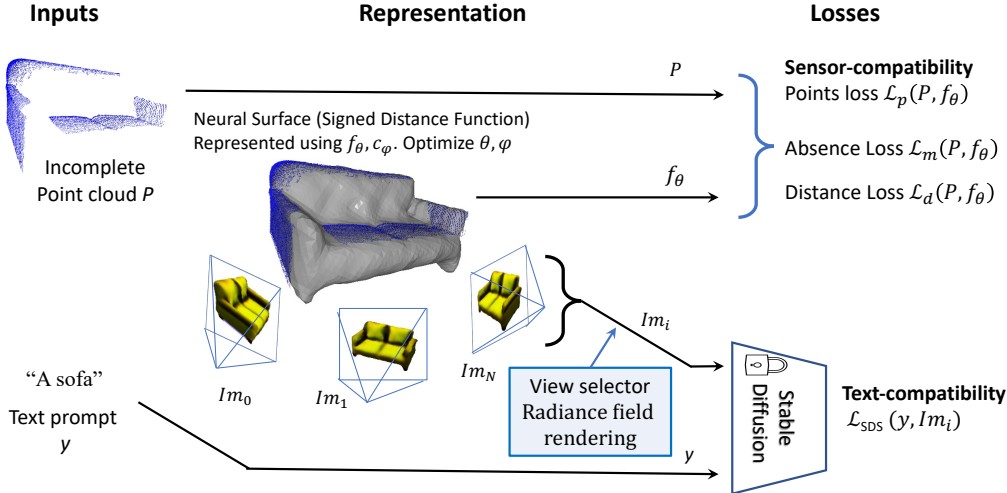

Figure 2: SDS-Complete optimizes two neural functions: A signed distance function $f_\theta$ representing the surface and a volumetric coloring function $\mathbf{c}_\varphi$. Together, $(\mathbf{c}_\varphi, f_\theta)$ define a radiance field, which is used to render novel image views $Im_0, \ldots Im_n$. The SDS loss is applied to the renderings to encourage them to be compatible with the input text $\mathbf{y}$. Three sensor-compatibility losses verify that the reconstructed surface is compatible with the sensor observations in various aspects.

**Inputs and components of our system.** The overall scheme for our method is depicted in Fig. 2. Two inputs are expected, (Fig. 2 left): a set of 3D input points $P = \{\mathbf{p}_1, \mathbf{p}_2, \ldots, \mathbf{p}_N\}$ measured relative to the sensor's location, and a text description embedding $\mathbf{y}$ of the incomplete object. As in previous methods [53, 56], we assume that the point cloud is segmented out from the original scan, namely, that all the points in $P$ belong to a single object.

Our task is to find the 3D surface (Fig. 2 center) of the complete object that is consistent with both the input points $P$ and the text prompt $\mathbf{y}$. For any given point cloud, our method optimizes for the complete object surface represented by a neural signed distance function $f_\theta : \mathbb{R}^3 \to \mathbb{R}$, and a neural color function $\mathbf{c}_\varphi : \mathbb{R}^3 \to \mathbb{R}^3$, where $\theta$ and $\varphi$ represent the learned parameters of the neural functions. As shown in [55], these two functions form a neural radiance field [33] and can be optimized using the rendered images of the 3D volumetric functions. More background details are given in the appendix.

## 3.2 Overview of the Optimization Process

Our main goal is to reconstruct a surface that is consistent with the partial input point cloud $P$, typically capturing only "one side" of the object. Clearly, constraining the surface to be consistent with the observed input point cloud is not sufficient for determining the surface on the "other side" of the object, and some prior knowledge should be used. Traditionally, such prior knowledge is learned by training a model over a large dataset of 3D shapes [53, 56]. Here, we instead use a pre-trained text-to-image diffusion model, applying an SDS loss [39] to rendered images of the object (Fig. 2 bottom). To ensure that we correctly reconstruct all sides of an input object, we use two types of compatibility losses: (1) **Sensor-compatibility losses (Fig. 2 top-right)**. (2) **Text-compatibility loss (Fig. 2 bottom-right)**. These losses are described below in more detail in Sec.3.3.

Applying and combining these compatibility losses is far from trivial and we now discuss several critical issues when using them. First, unlike DreamFusion [39], where each object is generated "from scratch", the *pose* of a generated object in our setup is determined by the input points $P$. As a result, camera positions must be sampled in a way that is compatible with the image prior, namely, that they render the object in natural poses.

Second, we find that simply rendering the object from the "other side" (unobserved side) and applying the SDS loss tends to generate content that is compatible with the input text but not with the input point cloud. To address this, we define a "curriculum" of sampling camera poses. We start from the known original viewpoint of the sensor that captured the points and gradually increase the range of views that we sample from around the original view (details in the appendix). With this sampling protocol, generated completions are continuously kept compatible with both the description text and the point cloud generated so far, until the entire object is completed successfully. As shown in our ablation study, this camera sampling protocol is key for producing high-quality object completions.

We next describe in more detail the compatibility losses and the camera-sampling process.

### 3.3 Training Losses

**Sensor-compatibility losses.** The sensor compatibility losses are used for constraining the output surface to be compatible with the input point cloud $P$. The surface is defined as the zero-level set of our optimized neural SDF $f_\theta$. Therefore, for constraining the surface to go through the point cloud, we encourage the function to be zero at these points.

$$\mathcal{L}_p = \frac{1}{N} \sum_{i=1}^{N} |f_\theta(\mathbf{p}_i)|. \tag{1}$$

We note that the points in $P$ are produced by a subset of sensor rays. Sensor rays that do not produce any points in $P$ define constraints on the boundary of the object. We use this to define an additional sensor-compatibility loss $\mathcal{L}_{\mathrm{m}}$. For each sensor ray $i$, we denote its opacity by $M_i \in \{0, 1\}$ where $M_i = 1$ if the ray produces a point in $P$ and $M_i = 0$ otherwise. We further denote its rendered opacity by $\tilde{M}_i \in [0, 1]$. The absence loss for the mask $M$ is defined by:

$$\mathcal{L}_{\mathrm{m}} = \frac{1}{K} \sum_{i=1}^{K} |M_i - \tilde{M}_i| \tag{2}$$

where $K$ is the number of sensor rays.

$\mathcal{L}_p$ constrains the location of the surface, but it does not constrain the sign of the values around the surface which indicates on which side the interior of the object is located. For that, we use the fact that each point in $P$ is measured relative to the sensor location. We denote the distance of each point from the sensor by $D_i$ and the rendered distance of the ray that produced this point by $\tilde{D}_i$. Our distance loss is defined by:

$$\mathcal{L}_{\mathrm{d}} = \frac{1}{N} \sum_{i=1}^{N} \left\| D_i - \tilde{D}_i \right\|^2 \quad, \tag{3}$$

**Text-compatibility loss.** We use a pre-trained text-to-image diffusion model, $\Phi$, to provide a semantic prior for predicting the unobserved parts, such that any rendered image of the object would be compatible with the input text embedding $\mathbf{y}$. To this end, we render random object views using our radiance field and apply the SDS loss with the input text embedding $\mathbf{y}$ to optimize $f_\theta$ and $\mathbf{c}_\varphi$ (Fig. 2, bottom-right). More details about the SDS loss are given in the appendix.

**Regularization losses.** To constrain $f_\theta$ to form a valid SDF, we apply the Eikonal loss regularization introduced in [18]:

$$\mathcal{L}_{eikonal} = \frac{1}{|P_{eik}|} \sum_{\mathbf{p} \in P_{eik}} | \|\nabla f_\theta(\mathbf{p}_i)\| - 1| \quad, \tag{4}$$

where $P_{eik}$ contains both $P$ and uniformly sampled points from the region of interest.

Finally, we use the known world plane to further prevent the surface from drifting below the ground:

$$\mathcal{L}_{plane} = \sum_{\mathbf{p} \in P_{\mathrm{uniform}}} \max(-f_\theta(\mathbf{p}), 0), \tag{5}$$

where $P_{\mathrm{uniform}}$ is a set of uniformly sampled $3D$ points below the plane in the region of interest.

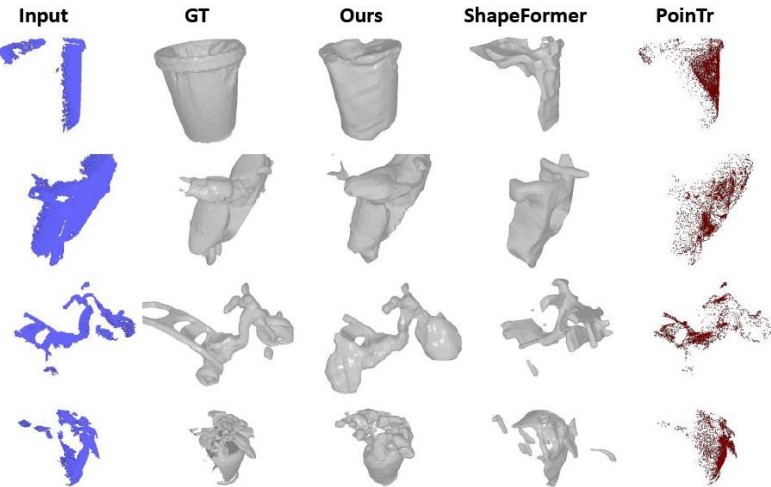

Figure 3: **Qualitative results for the Redwood dataset.** A qualitative comparison between our method and state-of-the-art methods. SDS-Complete produces more accurate completions.

Our total loss is:

$$\mathcal{L}_{\text{total}} = \delta_m \mathcal{L}_m + \delta_d \mathcal{L}_d + \delta_p \mathcal{L}_p + \delta_{eikonal} \mathcal{L}_{eikonal} + \delta_{plane} \mathcal{L}_{plane} + \mathcal{L}_{\text{SDS}}, \tag{6}$$

where $\delta_m, \delta_d, \delta_p, \delta_{eikonal}$ and $\delta_{plane}$ are the coefficients that define the weights of the different loss terms relative to the SDS loss and were selected by hyperparameters search. The same constant coefficients were used in all of our experiments. See appendix, for more implementation details.

### 3.4 Handling Camera Positions

As discussed above, the protocol for sampling camera views has a large impact on the quality of final completions. Let $C_0 = (R_0, \mathbf{t}_0)$ be the original camera-to-world pose of the sensor. To prevent rendering flipped or unrealistically rotated images of the object, we define the azimuth and elevation deviation from $C_0$ relative to the world plane. Specifically, let $\mathbf{n_l} \in \mathbb{S}^2$ be the normal to the world plane l, we define the azimuth rotation update to be $R_{\text{azimuth}} = \mathcal{R}(\mathbf{n_l}, \gamma_{\text{azimuth}})$, where $\mathcal{R}(\mathbf{n}, \gamma)$ is the Rodrigues' rotation formula for a rotation around the unit vector $\mathbf{n}$, with $\gamma$ degrees. Similarly, let $\mathbf{a}_0$ be the normalized principal axis direction of $C_0$, we define the elevation rotation update by $R_{\text{elevation}} = \mathcal{R}(\mathbf{n_l} \times \mathbf{a}_0, \gamma_{\text{elevation}})$. Assuming that the origin is located at the center of the object, an updated camera, $C_{\text{update}}$, for $\gamma_{\text{azimuth}}$ and $\gamma_{\text{elevation}}$ degrees, is given by:

$$C_{\text{update}} = (R_{\text{azimuth}} R_{\text{elevation}} R_0, R_{\text{azimuth}} R_{\text{elevation}} \mathbf{t}_0). \tag{7}$$

During training, we start by applying the SDS loss to rendered images from $C_0$ pose, and then we gradually increase the sampling range of the deviation angles until the entire object is covered. More implementation details are given in the appendix.

## 4 Experiments

**Datasets.** When considering evaluation, our primary goal is to evaluate our SDS-Complete and baseline method in real-world scenarios. This is in contrast to evaluating test splits from the synthetic datasets that were used for training the baseline methods. To achieve relevant evaluation datasets, we based the evaluation on partial real-world point clouds obtained from depth images and LiDAR scans.

For depth images, we used the Redwood dataset [12] that contains a diverse set of objects. We used depth images from 14 representative objects with ground truth $360°$ reconstructions which enable quantitative evaluation. We further tested our model on the KITTI LiDAR dataset [7, 17], which contains incomplete point clouds of objects in real-world scenes captured by LiDAR sensors. Both, our method and the baselines, require a segmented point cloud as inputs. Therefore as a preprocessing, we segmented out and centralized the main object from each scene (see more technical details in the

| Object | Shape Former | PoinTr | cGAN | Sinv | SDS-Complete | |
|---|---|---|---|---|---|---|
| | | | | | Full | Simple |
| Old chair | 23.2 | 34.1 | 33.2 | 36.7 | 19.3 | **18.9** |
| Outside chair | 25.9 | 29.6 | 42.8 | 28.7 | 22.6 | **22.4** |
| One lag table | 39.7 | 21.6 | 99.4 | 24.9 | 20.3 | **18.1** |
| Executive chair | 33.6 | 43.9 | 208 | **20.6** | 23.7 | **22.0** |
| Average | 30.6 | 32.3 | 95.8 | 27.7 | 21.5 | **20.4** |

Table 1: Chamfer loss (lower is better) for chair and table categories from the Redwood dataset. All baselines were trained on chairs and tables.

| Object | Shape Former | PoinTr | SDS-Complete | |
|---|---|---|---|---|
| | | | Full | Simple |
| Trash can | 136.4 | 137 | **36.4** | 43.1 |
| Plant in a vase | 60.8 | 41 | 29.5 | **27.4** |
| Vespa | 79.4 | 70.3 | 57.6 | **35.7** |
| Tricycle | 65.2 | 60.4 | **39** | 41.3 |
| Couch | 43.9 | 87.4 | **36.5** | 50.1 |
| Office trash | 68.8 | 49.7 | 20.5 | **18.7** |
| Plant in a vase 2 | 31.3 | 37.6 | 28.1 | **26.3** |
| Park trash can | 130 | 119.9 | 33.8 | **26.4** |
| Bench | **29** | 32.6 | 55.4 | 98 |
| Sofa | 106.6 | 129.3 | **40.6** | 43.2 |
| Average | 75.1 | 76.5 | **37.8** | 41.0 |

Table 2: Chamfer loss (lower is better) for general objects from the Redwood dataset. Most of the object categories are new for all methods and were not observed during training.

appendix). For each input object, the world's ground plane, $l \in \mathbb{P}^3$ that our method uses for camera sampling (Sec. 3.4) is extracted from each scene automatically by robust fitting. We also use the same world plane for the baseline methods as part of the point cloud to dataset alignment procedure that they require (see appendix for details).

**Baselines:** We compare our method to state-of-the-art point-cloud completion approaches: PoinTr [56], ShapeFormer [53], cGAN [50] and Sinv [58]. PoinTr and ShapeFormer trained one model on multiple classes; cGAN and Sinv trained per-class models for chairs and tables classes.

### 4.1 Results on the Redwood dataset

As our quality metric, we measure the Chamfer distances in millimeters to quantify the dissimilarity between the generated completions and their corresponding ground-truth shapes. Results are presented in Tables 1 and 2. Table 1 compares all methods on objects from two categories, tables and chairs; Table 2 compares our method to PoinTr [56] and ShapeFormer [53], on the remaining Redwood objects. Both tables demonstrate that SDS-Complete obtains state-of-the-art results on the task of point cloud completion from real point clouds.

In addition to the full variant discussed above, we also show results with a simplified model we name *"SDS-Complete Simple"*. This simplified variant omits the RGB network $c_\phi$, and uses gray-shaded rendering images of the optimized geometry for the SDS-loss. It is more efficient but has a higher average error. We further show qualitative results for general objects in Fig. 3. Qualitative results for chairs and tables are presented in the appendix.

**Completion with various text descriptions.** Our approach operates by combining a partial input point cloud with a text description that guides the model when completing missing parts of the object.

We tested the effect of changing the text prompt while keeping the same input point cloud. Figure 4 shows results for completing Redwood's scan "08754"(Teapot) of a partially captured teapot (Fig. 1). Completing the point cloud with other text descriptions demonstrates how the text controls the shape.

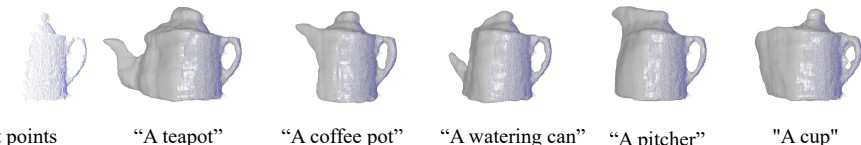

| Input points | "A teapot" | "A coffee pot" | "A watering can" | "A pitcher" | "A cup" |

Figure 4: **The effect of text prompt**. Completion of the same input, with different text descriptions. Results obtained with our method for the partial point cloud of scan "08754". While the handle and the top part of the object are constrained by the input point cloud, the model completes the other side of the object according to the input text.

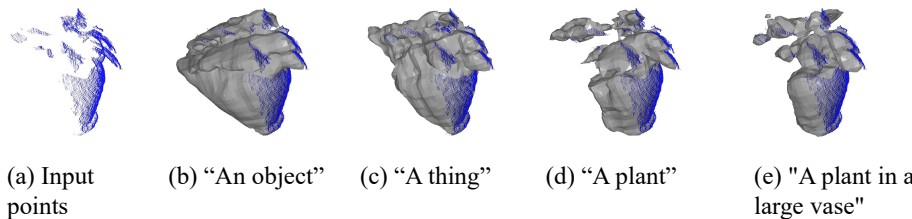

| (a) Input points | (b) "An object" | (c) "A thing" | (d) "A plant" | (e) "A plant in a large vase" |

Figure 5: **Generic vs specific text prompts: qualitative results.** Results are shown for reconstructing scan "06127" (Plant in a vase) from the Redwood dataset. **(a)** The input point cloud. **(b, c)** Completion using two generic texts. Completion quality is poor. **(d)** Completion using the object class name. **(e)** Completion using a detailed textual description.

**Generic vs specific Text Prompts.** To evaluate the contribution of selecting an appropriate text prompt per object, we repeated reconstruction experiments of the 14 objects that we evaluate in Tables 1 and 2, but varied the text prompts. Specifically, we used three levels of description specificity. First, for a fully generic prompt (class agnostic), we tested two alternatives: "An object", and "A thing". Second, we used the class name as the prompt ("A <class name>"). Finally, we used a more detailed description. The full-text prompts are given in the supplementary.

Table 3 shows the Chamfer distances between our reconstruction and the ground truth for all prompts. Using generic text yields inferior reconstructions. Adding specific details did not provide a significant improvement over using the class name. A qualitative comparison is shown in Fig. 5.

**Ablation study.** To evaluate the contribution of each component of our method we present qualitative and quantitative ablation studies in Fig. 6. As can be seen, without the SDS loss, our model has no understanding of object characteristics like the fact that the chair has four legs and a straight back-side. Without the SDF representation, it is not possible to apply the point cloud constraints directly on the surface which results in an inferior ability to follow the partial input. Finally, it can be seen that our camera sampling "curriculum" that is described in Sec. 3.4, improves the completion compared to a random camera sampling ("Naïve camera sampling") by preserving the consistency of the generated content with the existing sensor measurements and by verifying that the diffusion model does not see any "unnatural" pose of the object.

## 4.2   Results on the KITTI Dataset

We compare our method to ShapeFormer [53] and PoinTr [56] on a subset of 15 real object scans from the Semantic KITTI Dataset [8] which consists of 5 cars, 5 motorcycles, and 5 trucks. We present qualitative comparisons in Fig. 7. Notably, our method shows better completion results, particularly with motorcycle objects which are less frequent in the Shapenet dataset.

**User Study.** We conducted a user study to evaluate the various methods on the KITTI dataset. Specifically, we gathered a group of 11 participants to rank the quality of each completed surface and its faithfulness to the input partial point cloud. For each object, the participants were given three anonymous shapes produced by the three methods: SDS-Complete, ShapeFormer [53], and PoinTr

| Text Prompt | "An object" | "A thing" | "A <class name>" | Full text |
|---|---|---|---|---|
| Chamfer | 52.0 | 52.7 | 33.8 | 33.1 |

Table 3: **Generic vs specific text prompts: Chamfer distances (lower is better)**. Columns 1 and 2: two generic configurations where a global text is used for all objects. Column 3: Only the class name is used e.g. both "executive chair" and "outside chair" are reconstructed with the text "A chair". Column 4: The results of our method with the full text prompts (provided in the supplementary).

| Input points | Naïve camera sampling | No SDS loss | No SDF representation | Full method |
|---|---|---|---|---|

| | Naïve camera sampling | No SDS loss | No SDF representation | Full method |
|---|---|---|---|---|
| **Quantitative** (Avg. Chamfer) ↓ | 42.3 | 46.9 | 75.0 | **33.1** |

Figure 6: **Ablation study**. We demonstrate the contribution of each part of our method. Naïve camera sampling: running without our camera handling that is described in Sec. 3.4. No SDS loss: using all losses but the SDS loss. No SDF representation: running with a density function as in [33]. Below, we **quantitatively** compare the average Chamfer distance over the evaluated 14 Redwood scans. We show extended ablations in the appendix.

[56]. While the outputs of SDS-Complete and ShapeFormer are surfaces, PoinTr only outputs a point cloud. Therefore, we applied Screened Poisson Surface Reconstruction [24] to each output of PoinTr to base the user study comparisons on surface representations. The participants were instructed to choose the best shape, while the order of the methods was shuffled for each object. The best completion method for each input case is selected by the majority vote. The results of the user study are presented in Table 4, showing that our method got the highest number of wins (14 out of 15).

**Quantitative evaluation on KITTY.** For a quantitative metric, we followed PCN [57] and calculated the Minimal Matching Distance (MMD). MMD is the Chamfer Distance (CD) between the output surface and the surface from ShapeNet that is closest to the input point cloud in terms of CD. We calculated this metric on the surfaces that were evaluated in our user study from two categories: car and motorcycle. These are the only categories that have associated Shapenet subsets which is a necessary condition for calculating the MMD metric. The mean MMD over the motorcycle and car shapes are presented in Table 4, showing that our approach improves over the baselines.

We further computed the CLIP R-Precision metric [39] on all of our evaluated KITTI categories: "car", "truck" and "motorcycle". This metric checks the accuracy of classifying a rendered image by choosing the class that maximizes the cosine similarity score between the image and the text: "a rendering of a <class name>" among all classes. We evaluated the output geometries of the different methods, each rendered from 360 degrees with azimuth gaps of 2 degrees (180 images for each surface). We report the mean accuracy in Tab. 4. Here again, our approach is substantially better.

### 4.3 Limitations

As in previous work ([53]), for extracting point clouds from depth images, we need the internal parameters of the depth camera. Our test time optimization method is slow compared to the feedfor-

| | Shape Former | PoinTr | SDS-Complete (ours) |
|---|---|---|---|
| User-study wins [%] ↑ | 7.0 | 0.0 | **93** |
| Average MMD ↓ | 0.036 | 0.052 | **0.027** |
| CLIP R-Precision ↑ | 0.61 | 0.43 | **0.76** |

Table 4: **Quantitative results on KITTI**. User-study wins refer to the percentage of cases in which a method was selected by raters as the best method. Our method outperforms the baseline methods in all 3 metrics.

| Input | SDS-Complete | ShapeFormer | PoinTr |

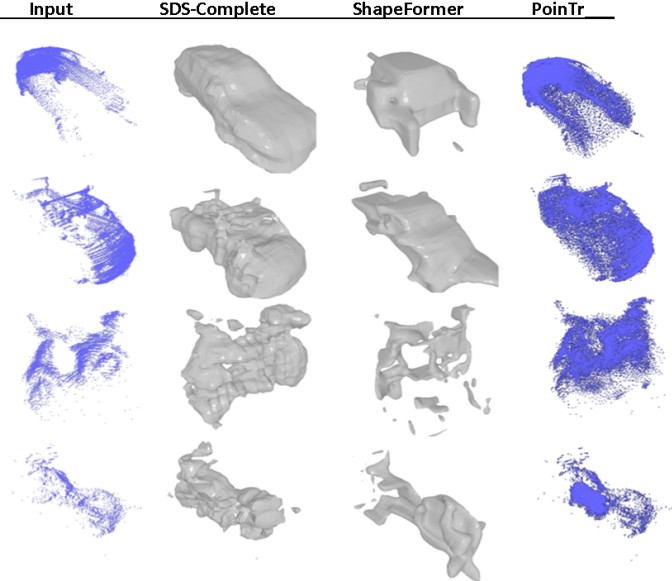

Figure 7: Qualitative completion results on the KITTI dataset.

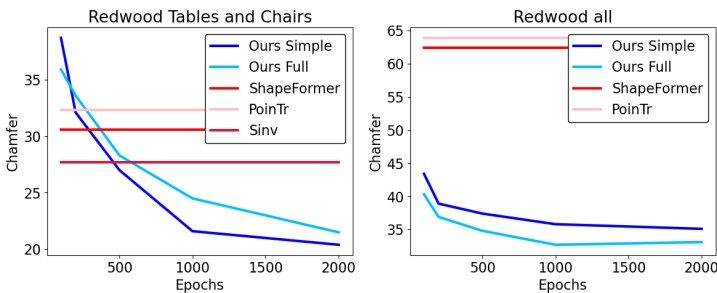

Figure 8: **Shorter Training.** The Chamfer error when running our method for fewer epochs. For reference, we also include the (inference time) numbers of the baselines. Left: our convergence on the table and chair categories. Right: our convergence when considering all 14 Redwood models.

ward baseline methods that were pre-trained on large datasets of 3D shapes. The major factor for the running time is the SDS-loss which needs many sampling views. The number of training epochs can be shortened by reducing the number of iterations. In Tab. 8 we show the effect of reducing our runtime. As we show, our method outperforms the baselines in terms of average accuracy after 5% of the training time and keeps improving when more training time is given. In the appendix, we show failure cases of our method and list additional implementation details.

## 5   Conclusions

We presented SDS-Complete, a novel test time optimization approach for 3D completion using a text-to-2D pre-trained model. For handling point cloud inputs, we incorporated an SDF representation and constrained the surface to lie on the input points. We successfully applied the SDS loss on images rendered from novel views and completed the missing part of the object by aligning the images with an input textual description. By handling the camera sampling carefully we maintained the consistency of the completed part with the input captured part. This enabled us to produce superior results even on previously unconsidered objects for completion. Future work includes improving running times by combining recent NeRF techniques (e.g. [35]), and supporting scene completion from incomplete point clouds of multiple objects.

# 6 Acknowledgments

We thank Lior Yariv, Dolev Ofri, Or Perel and Haggai Maron for their insightful comments. We thank Lior Bracha and Chen Tessler for helping with the user study. This work was funded by a grant to GC from the Israel Science Foundation (ISF 737/2018), and by an equipment grant to GC and Bar-Ilan University from the Israel Science Foundation (ISF 2332/18). OR is supported by a PhD fellowship from Bar-Ilan data science institute (BIU DSI).

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

# A Implementation Details

**Running Time**   We run our method for 2000 epochs, where each epoch uses 100 iterations. That takes about 1950 and 1380 minutes for Redwood and KITTI scans respectively. The optimization time for our simplified model takes about 1256 and 924 minutes for Redwood and KITTI scans respectively. All times are measured when running our method on NVIDIA RTX A6000. We note that many scans need much fewer iterations for converging (see Fig. 8), but to complete the fine details, e.g. the chair's legs, many iterations are needed.

**Network Architecture**   For our optimized coloring function $c_\varphi$, we use 4 linear layers with 96 channels, where the two intermediate ones are incorporated in ResNet [21] blocks, as implemented by [47], with SiLU activations [22]. For the SDF network $f_\theta$, we use 4 linear layers with 96 channels and ReLU activations. $f_\theta$ is initialized to a sphere [5] with radius lengths of $0.5$ and $0.9$ for Redwood and KITTI scans respectively. For both, $c_\varphi$ and $f_\theta$ we use Positional Encoding with 6 levels [33]. For extracting density from $f_\theta$ (Equations (12) and (13)) we use $\alpha = 100, \beta = 10^{-3}$.

**SDS-loss Implementation Details**   We base our code on the implementation of [47]. During training, for each iteration, we randomly sample a background color, to prevent the model from "changing" the geometry by just coloring it with the background color. For Redwood cases, we render $80 \times 80$ images for the SDS-loss using the sampled camera and the known internal parameters of the sensor. For KITTI, at initialization time we first project the object's LiDAR points to a 2D spherical projection [8], with height and width of 64 and 1024 pixels respectively. We use the projected 2D mask to select the 2D bounding box area of 64 pixels height, where the width is determined by the min and max horizontal coordinates of the object $\pm 5$ pixels. The LiDAR rays that define this selected bounding box are used to render the object during training, where a novel camera pose is defined by rotating these rays around the object's centroid. As a text-to-image diffusion model we use Stable Diffusion v2 [43].

**Training Details**   We optimize the networks using the Adam optimizer [26] with a learning rate $10^{-4}$. The coefficients for our loss for all the experiments are $\delta_m = 10^5, \delta_d = 10^5, \delta_p = 10^5, \delta_{eikonal} = 10^4, \delta_{plane} = 10^5$. At each iteration we sample 1000 uniform points for $\mathcal{L}_{plane}$ and $\mathcal{L}_{eikonal}$. For $\mathcal{L}_m, \mathcal{L}_d$, at each iteration, we randomly sample 2000 pixels for Redwood cases, whereas for KITTI, we render the entire bounding box.

**Camera Sampling**   As described in Sec.3.4, during training, we start by applying the SDS loss on the rendered image from $C_0$ pose, and then we gradually increase the sampling range of the deviation angles until the entire object is covered. In more detail, we gradually increase the sampling range of the azimuth angles: $\gamma_{azimuth} \sim \mathcal{U}(-\nu, \nu)$, starting from $\nu = 0$ to $\nu = 180$. Specifically, we set $\nu = 30, 45, 60, 90, 180$ at epochs $20, 50, 80, 100, 120$ respectively. $\gamma_{elevation}$ is set to 0 for 20 epochs and then uniformly sampled according to: $\gamma_{elevation} \sim \mathcal{U}(-\xi_0, 0)$ for Redwood scans, where $\xi_0$ is the elevation of $C_0$ from the plane l in degrees. For KITTI scans (after epoch 20) we use $\gamma_{elevation} \sim \mathcal{U}(-\xi_0, \xi_0)$ since the original viewpoint is usually low, and we also scale the distance from the source to the object uniformly by $\sim \mathcal{U}(1, 2)$ after epoch 20. As in [39], we augment the input text according to the viewing direction, with a text that describes the viewpoint orientation. Specifically, as in [47] we use "*, front view", "*, side view", "*, back view", "*, overhead view" and "*, bottom view", where * denotes the input text. Unlike [39], the orientation of the object is determined by the input points. Therefore, we use an extra input from the user of $\gamma_{0_{azimuth}}$, which explains the original viewpoint, e.g. $\gamma_{0_{azimuth}} = 90$ if the object is viewed from the side. Then, during training, we use $\gamma_{0_{azimuth}}$ and $\gamma_{azimuth}$ to calculate the azimuth with respect to the object, and $\gamma_{elevation}$ to compute the elevation with respect to the plane l. These orientations are used to augment the text with the corresponding view direction description. In our simplified version ("SDS-Complete Simple") we omit the text augmentation and hence we do not need this extra input from the user of $\gamma_{0_{azimuth}}$.

**Object Centralization**   Given the input points we centralize them at the origin. This is done in general by subtracting their center of mass. When the object's largest dimension is aligned with the viewing axis, the center of mass is usually biased toward the camera. To handle this, we extract an oriented 3D bounding box for the input points and measure the ratio between the largest distance to the smallest distance from the center of mass to any bounding box point. If this ratio is above $1.7$ we

use the bounding box center as our centroid instead of using the center of mass. In the KITTI dataset, which mostly includes non-isotropic objects, we always use the bounding box center as our centroid. We then scale the points such that the largest point norm is $0.5$.

**Baseline Runnings** For running the baseline methods, we tried to locate the input points as much as possible according to the method's expectations to prevent them from failing. This includes using our knowledge about the world plane $l$ and the object orientation with respect to the camera $\gamma_{0_{\text{azimuth}}}$. For ShapeFormer, each time we took the best shape out of the 5 that it outputs.

**Data Processing** For the Redwood dataset, we segmented out the foreground object manually. For the evaluation only, as a preprocessing, we manually aligned the GT scan with the partial point cloud and applied ICP for refinement [4]. Each KITTI scan that we used, is the aggregation of 5 timestamps. The segmentation map for KITTI is given by [8]. For both, KITTI and Redwood datasets and for each scan, the plane $l$ is segmented out from the original point cloud using RANSAC [20].

# B  Preliminaries

## B.1  Volume Rendering

**Neural Radiance Field** A neural radiance field [33] is a pair of two functions: $\sigma : \mathbb{R}^3 \to \mathbb{R}^+$ and $\mathbf{c} : (\mathbb{R}^3, \mathbb{S}^2) \to \mathbb{R}^3$, each represented by a Multilayer Perceptron (MLP). The function $\sigma$ maps a 3D point $\mathbf{x} \in \mathbb{R}^3$ into a density value, and the function $\mathbf{c}$ maps a 3D point $\mathbf{x}$ and a view direction $\mathbf{v} \in \mathbb{S}^2$ into an RGB color. A neural radiance field can represent the geometric and appearance properties of a 3D object and is used as a differentiable renderer of 2D images from the 3D scene. Let $I$ be an image with a camera center $\mathbf{t} \in \mathbb{R}^3$, the pixel coordinate $\mathbf{u} = (u, v)^T \in \mathbb{R}^2$ is backprojected into a 3D ray $r_{\mathbf{u}}$, starting at $\mathbf{t}$ and going through the pixel $\mathbf{u}$ with a direction $\mathbf{v} \in \mathbb{S}^2$. Let $\mu_1, \mu_2, \ldots, \mu_{N_r}$ be sample distances from $\mathbf{t}$ on the ray $r_{\mathbf{u}}$, then the densities and colors of the radiance field are alpha composited from the camera center through the ray. The RGB image color $I(u, v)$ is calculated by:

$$I(u, v) = \sum_{i=1}^{N_r} w_i \mathbf{c}(\mathbf{t} + \mu_i \mathbf{v}, \mathbf{v}) \tag{8}$$

where $w_i = \alpha_i \prod_{j<i}(1 - \alpha_j)$ is the color contribution of the $i^{th}$ segment to the rendered pixel, and $\alpha_i = 1 - \exp\left(-\sigma(\mathbf{t} + \mu_i \mathbf{v})(\mu_{i+1} - \mu_i)\right)$ is the opacity of segment $i$. Eq. (8) is differentiable with respect to the learned parameters of $\mathbf{c}$ and $\sigma$ and therefore, is used to train the neural radiance field. Let $\bar{I}$ be the ground truth image, then the MSE loss is used to train the neural radiance field:

$$\mathcal{L}_{MSE} = \frac{1}{n} \sum_{i=1}^{n} \left\| I(\mathbf{u}_i) - \bar{I}(\mathbf{u}_i) \right\|^2 \tag{9}$$

where $n$ is the number of pixels in the batch.

**Volume Rendering of Neural Implicit Surfaces** While the neural radiance field shows impressive performances in synthesizing novel views, extracting object geometries from a trained radiance field is not trivial. Defining the surface by simply thresholding the density $\sigma$ results in noisy and inaccurate geometry. We adopt the solution proposed by [55]. Let $\Omega \subset \mathbb{R}^3$ be the space occupied by the object, and $\mathcal{M}$ denotes the boundary of the surface. Then the SDF $f : \mathbb{R}^3 \to \mathbb{R}$ is defined by

$$f(\mathbf{x}) = (-1)^{\mathbf{1}_\Omega(\mathbf{x})} \min_{\mathbf{y} \in \mathcal{M}} \|\mathbf{x} - \mathbf{y}\| \tag{10}$$

where $\mathbf{1}_\Omega(\mathbf{x}) = \begin{cases} 1 & \mathbf{x} \in \Omega \\ 0 & \text{otherwise} \end{cases}$ . Given $f$, the surface $\mathcal{M}$ is defined by its zero level set, i.e.

$$\mathcal{M} = \{\mathbf{x} \in \mathbb{R}^3 : f(\mathbf{x}) = 0\} \tag{11}$$

A signed distance function can be utilized for defining a neural radiance field density. Let $\mathbf{x} \in \mathbb{R}^3$ and $f : \mathbb{R}^3 \to \mathbb{R}$ be a 3D point and an SDF respectively, the density $\sigma(\mathbf{x})$ is defined by:

$$\sigma(\mathbf{x}) = \alpha \Psi_\beta(-f(\mathbf{x})) \tag{12}$$

where $\Psi_\beta(s)$ is the Cumulative Distribution Function (CDF) of the Laplace distribution with zero mean and $\beta$ scale:

$$\Psi_\beta(s) = \begin{cases} \frac{1}{2}\exp\left(\frac{s}{\beta}\right) & s \leq 0 \\ 1 - \frac{1}{2}\exp\left(-\frac{s}{\beta}\right) & s > 0 \end{cases} \tag{13}$$

and $\alpha$ and $\beta$ are parameters that can be learned during training (in our case, we set them to be constant). It is then possible to train a neural radiance field, defined by the SDF $f$ and the neural color function $\mathbf{c}$, using the loss function defined by Eq. (9).

### B.2 Score Distillation Sampling (SDS)

**Diffusion Models** A diffusion model [36, 45, 43] generates image samples from a Gaussian noise image, by inverting the process of gradually adding noise to an image. This process is defined as follows: at time $t = 1, \ldots, T$, a Gaussian noise $\epsilon \sim \mathcal{N}(\mathbf{0}, I)$ is added to the image: $I_t = \sqrt{\bar{\alpha}_t}I + \sqrt{1 - \bar{\alpha}_t}\epsilon$, where $\bar{\alpha}_t = \prod_{i=1}^t \alpha_i$, $\alpha_t = 1 - \beta_t$ and $\beta_t \in (0, 1)$ defines the amount of added noise. A denoising neural network $\hat{\epsilon} = \Phi(I_t; t)$ is trained to predict the added noise $\hat{\epsilon}$ given the noisy image $I_t$ and the noise level $t$. The diffusion models are trained on large image collections $\mathcal{C}$ for minimizing the loss

$$\mathcal{L}_{\text{D}} = E_{I \in \mathcal{C}}\left[\left\|\Phi(\sqrt{\bar{\alpha}_t}I + \sqrt{1 - \bar{\alpha}_t}\epsilon; t) - \epsilon\right\|^2\right] \tag{14}$$

Given a pretrained $\Phi$, an image sample is generated by sampling a Gaussian noise image $I_T \sim \mathcal{N}(0, I)$ and gradually denoising it using $\Phi$.

Diffusion models can be extended to be conditioned on additional inputs. Text-to-image diffusion models [43] condition $\Phi$ on a textual prompt embedding input $\mathbf{y}$, and train $\Phi(I_t; t, y)$. Therefore, they can generate images given text and random Gaussian noise.

DreamFusion[39] uses a pretrained, and fixed, text condition diffusion model $\Phi(I_t; t, y)$ and uses it to train a NeRF model from scratch, given a textural description embedding $\mathbf{y}_0$. In each iteration, a camera view is sampled and used to render an image $I_0$ from the NeRF model. $I_0$ is differentiable with respect to the learned parameters of the NeRF model ($\theta_{\text{NeRF}}$), and used as an input to $\Phi(I_0; t, y)$. The Score Distillation Sampling (SDS) loss is then applied:

$$\nabla_{\theta_{\text{NeRF}}}\mathcal{L}_{\text{SDS}}(I_0) = E_{t,\epsilon}\left[(w(t)\Phi(\sqrt{\bar{\alpha}_t}I_0 + \sqrt{1 - \bar{\alpha}_t}\epsilon; t, y_0) - \epsilon)\nabla_{\theta_{\text{NeRF}}}I_0\right] \tag{15}$$

Note that $\nabla_{\theta_{\text{NeRF}}}\mathcal{L}_{\text{SDS}}$ is the gradient with respect to $\theta_{\text{NeRF}}$ of Eq. (14), where the Jacobian of $\Phi$ is omitted for stability and efficiency. Intuitively, if $I_0$ looks like a natural image, and is compatible with $\mathbf{y}_0$, then the pretrained diffusion model predicts the added noise successfully, resulting in low values for $\mathcal{L}_{\text{D}}$. By updating the NeRF's weights according to Eq. (15), $\mathcal{L}_{\text{D}}$ is reduced, and as a result, the rendered images become more compatible with $\mathbf{y}_0$.

## C Limitations Examples

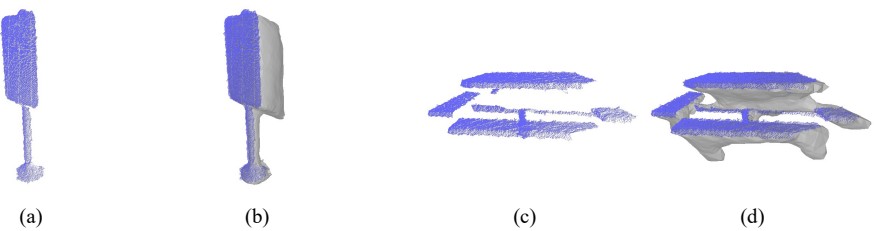

|     |     |     |     |
| :-: | :-: | :-: | :-: |
| (a) | (b) | (c) | (d) |

Figure 9: **Failure cases of our method.** (a),(b) Input points and surface completion respectively, for Redwood scan "05492" (standing sign). (c),(d) Input points and surface completion respectively for Redwood scan "01373" (picnic table).

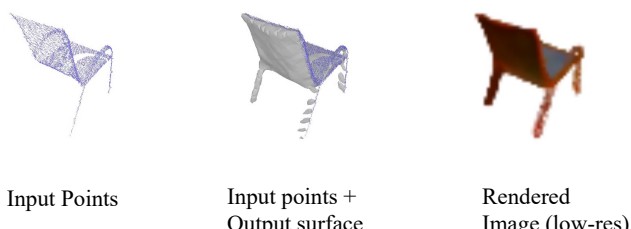

| Input Points | Input points +
Output surface | Rendered
Image (low-res) |

Figure 10: **Intermittent patterns in the legs of the chair.** The supervision signal for the legs, coming from the thin lines of input points (left) is not enough for reconstructing the chair's legs correctly (middle). The SDS loss sees valid rendered images in this case and therefore does not fix it (right).

**Failure Cases**   Fig. 9 shows failure examples. In general, our method does not reconstruct well thin surfaces. We hypothesize that the initialization of the SDF to sphere [5], prevents the model from minimizing the occluded part at early training stages. Then, the SDS loss usually tries to paint this redundant content according to the text prompt, instead of removing it. Different initializations to the SDF, or other regularizations, need to be explored and left as future work.

**Intermittent patterns**   Fig. 10 demonstrates a case where the thin structure of the input points is too weak to constrain the surface of the outside chair. Unfortunately, the SDS loss does not help in making the surface thicker since the rendering of the thin surface looks valid. This is due to our rendering process that uses VOLSDF which defines a smooth mapping from surface to density (Equation (12)), and since the entire leg has SDF values that are close to 0, the legs get densities that produce valid low-resolution renderings.

## D   Additional Results

**Reconstruction with incorrect prompts.**   We further check the sensitivity of our method to wrong text prompts. Specifically, we used the text: "A table" for a chair, and the text "A chair" for a table. The visualizations are presented in Fig. 11.

**Supplementary Redwood Results**   Qualitative results for table and chairs categories from the Redwood dataset are presented in Fig. 14. Additional qualitative comparisons for other objects from the Redwood dataset are presented in Fig.12. We can see that our method completes the shapes better than the baselines.

We applied our method to additional Redwood cases of various object types with no available ground truth. Qualitative results, including RGB renderings, are shown in Fig. 13.

**Video Results**   We attach to the supplementary folder, $360^o$ video visualizations of our reconstructed objects for both, KITTI and Redwood datasets.

**Extended Ablation**   We show an extended ablation study in Tab. 5 that shows a quantitative evaluation of the importance of the depth and mask losses. We further show separate numbers for the table and chair categories and the rest of the categories.

**Reduced input points**   We tested our "Full" method on the Redwood dataset with 50% and 10% of the original input points, on the evaluation set from the Redwood dataset. The locations of the removed points were selected randomly. We found that there is no significant difference in the results (up to 1.5 mm difference in terms of average Chamfer distance) when excluding the "Bench" case. For the "Bench" case there is no significant difference when using 50% of the points, but when using 10% of the points we see that the Chamfer distance increases from 55.4 to 149. These results indicate that in general, our method is robust to the number of input points.

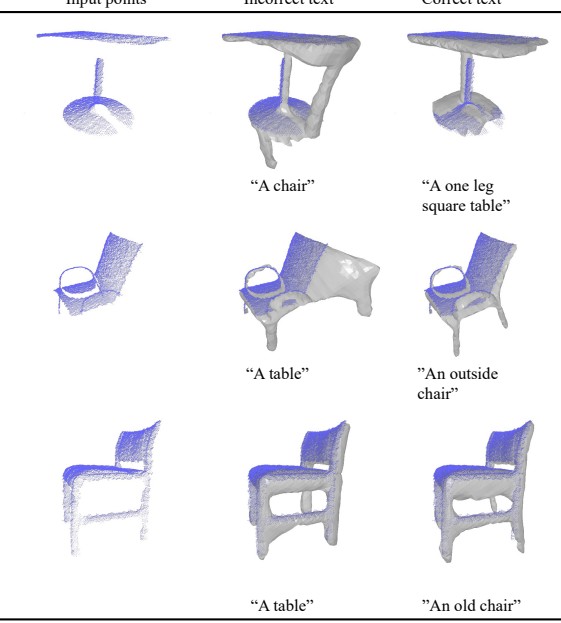

| Input points | Incorrect text | Correct text |
|---|---|---|
| | "A chair" | "A one leg square table" |
| | "A table" | "An outside chair" |
| | "A table" | "An old chair" |

Figure 11: The effect of incorrect textual descriptions. Each row corresponds to a different object. **Left:** The partial scans that are given as input to our model. **Middle:** Completion performed using incorrect text descriptions. **Right:** the completion results of our method with our final text prompts. In the first two rows, the completion is inferior when given the wrong text. In the bottom row, even with an incorrect text ("A table") the model still completes the chair correctly. This is because the input provides strong constraints. To make the shape more similar to a table, the method still needs to reconstruct the missing leg.

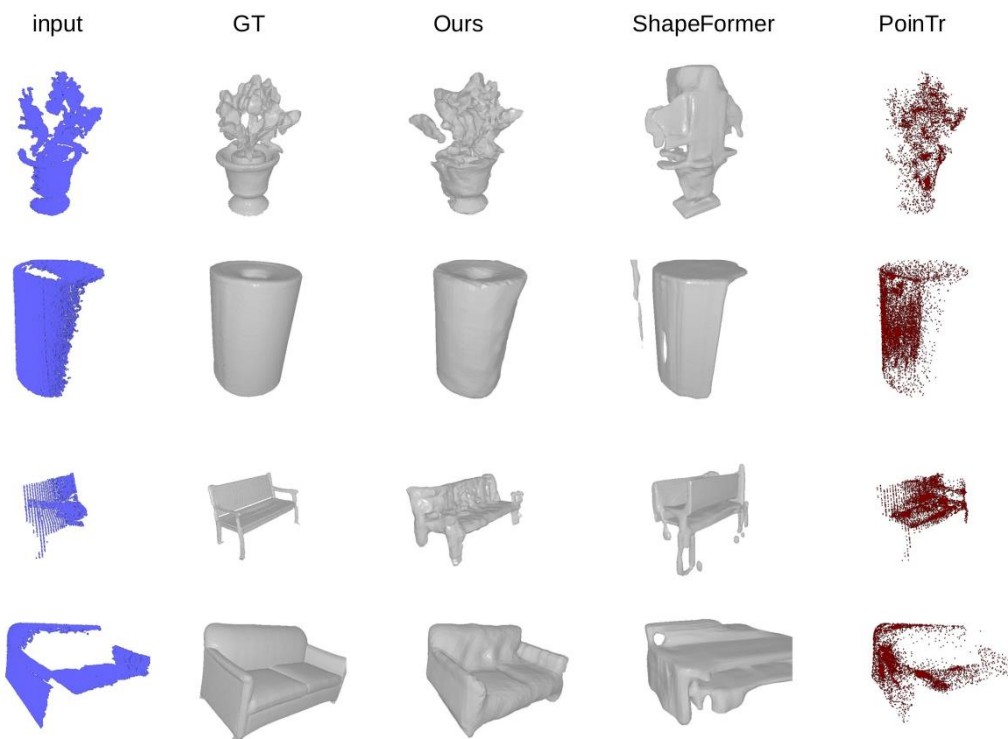

Figure 12: Additional qualitative comparisons on the Redwood dataset.

|                  | Naïve camera sampling | No SDS | No SDF | No Depth | No Mask | Full |
|------------------|:---------------------:|:------:|:------:|:--------:|:-------:|:----:|
| Redwood T & C    | 36.6                  | 44.0   | 59.9   | 28.0     | 58.0    | 21.5 |
| Redwood general  | 44.6                  | 48.1   | 81.0   | 45.1     | 99.1    | 37.8 |
| Redwood all      | 42.3                  | 46.9   | 75.0   | 40.2     | 87.4    | 33.1 |

Table 5: An ablation study for demonstrating the contribution of each part of our method. Naïve camera sampling: running without our camera handling that is described in Sec. 3.4. No SDS loss: using all losses but the SDS loss. No SDF representation: running with a density function as in [34]. No Depth: using all losses but the Depth loss. No Mask: using all losses but the Mask loss. The 3 rows present the average numbers for the table and chair categories (Redwood T & C), for all categories except table and chairs (Redwood general), and for the complete evaluation set (Redwood All).

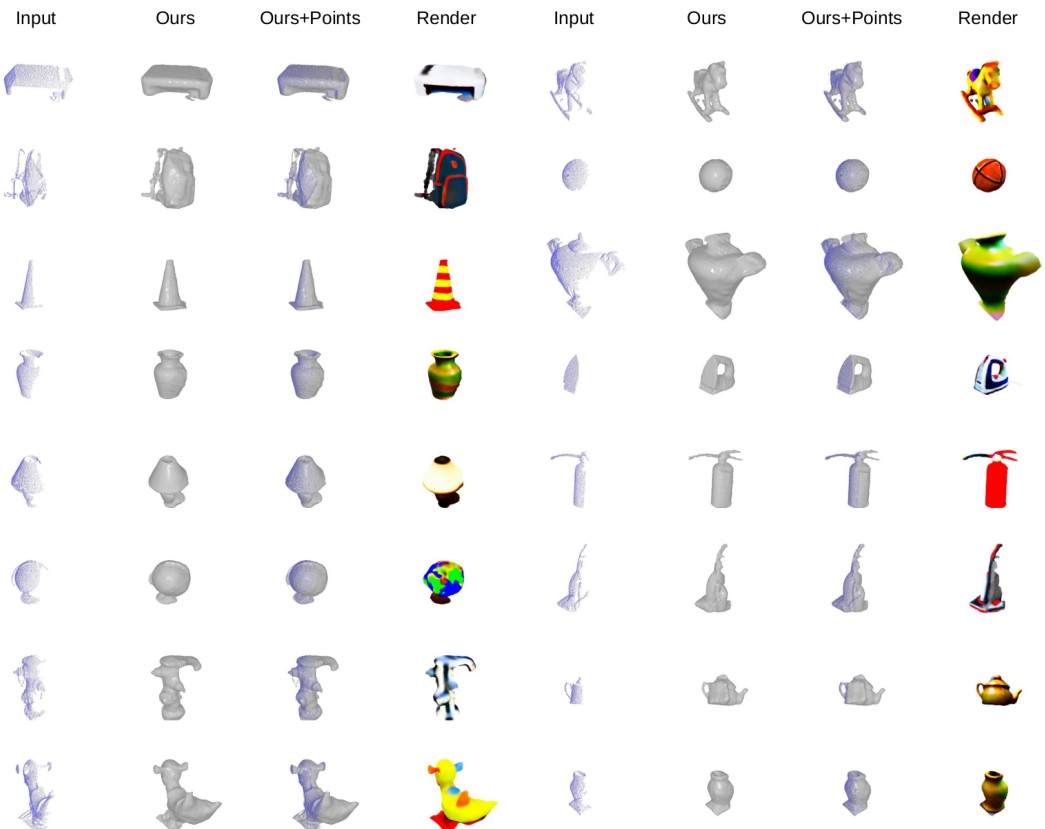

Figure 13: Qualitative outputs of our method, when applied on Redwood cases with no available $360^o$ GT scans for quantitative comparison. The figure is arranged as 2 columns of different objects, where for each object we show (from left to right): the input point cloud, the completed surface, the completed surface together with the input points, and image rendering of our optimized coloring function **c**.

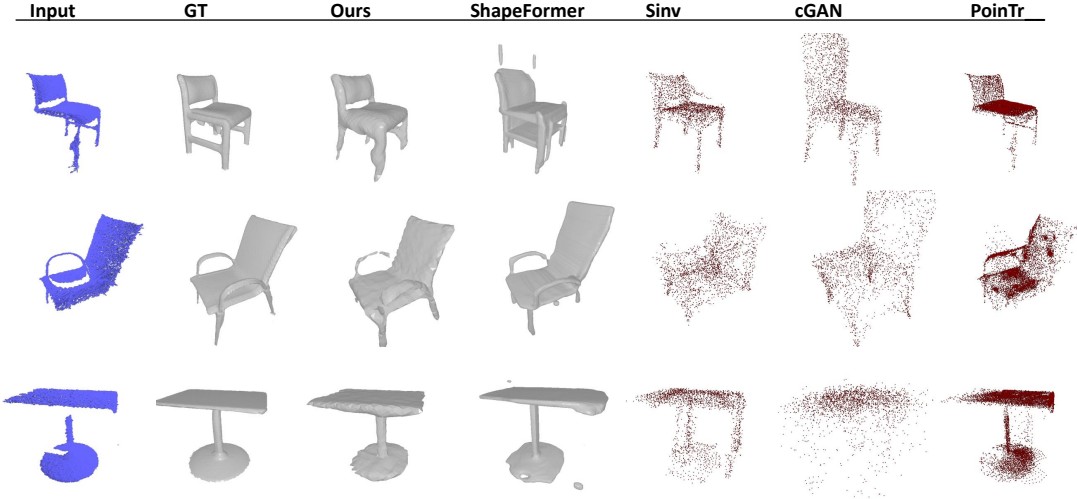

Figure 14: **Qualitative results for the table and chair categories from the Redwood dataset.** Red represents methods that output point clouds.

