# OpenReview forum: "Point Cloud Completion with Pretrained Text-to-Image Diffusion Models"
_NeurIPS.cc/2023/Conference — NeurIPS 2023 poster_

### Official Review · Reviewer_MGrG · 2023-06-23

**Soundness:** 2 fair
**Presentation:** 2 fair
**Contribution:** 2 fair
**Rating:** 4
**Confidence:** 3

**Summary:**

This paper proposes a optimization method for point cloud completion by leveraging a 2D diffusion model. By constructing a neural SDF field, the proposed method make sure that the represented object matches the input partial point cloud, while the rendering images fit a pre-trained Stable Diffusion by the guidance of SDS loss. The results show that the proposed method does not need large dataset like previous works do.

**Strengths:**

1. The proposed method does not need large dataset for training class specific completion model, making it more practical.


**Weaknesses:**

1. Looks like the proposed method simply adds a new constraint (matching surface with input partial point cloud) on DreamFusion-like structure and the rest are just existing modules and techniques. Although I believe this is technically not easy, the novelty is still limited.
2. In Figure 1, the author mentioned in distribution object and out-of-distribution object. However, this grouping method is meaningless for the proposed method since a pre-trained Stable Diffusion is utilized. The real out-of-distribution object for this model should be an object that even stable diffusion cannot generate a reasonable image for it, and clearly this situation is fatal for the proposed method. I think the author should not emphasize in or out-of-distribution here.
3. The comparisons in this paper seem to be unfair since previous works are trained under a totally different setting. The authors should show results of the same dataset (ShapeNet) with the previous works, which I believe existing works would outperform the proposed method.


**Questions:**

1. I notice that you still have a color MLP head for the object even though your task is just point cloud completion. Is it because rendering colored images is more favorable for Stable Diffusion? Is it possible to take off the color head and directly assign a fixed color to your neural SDF and add the corresponding color to the input text for Stable Diffusion to reduce the complexity of your neural SDF?
2. Please respond to the weakness part.

**Limitations:**

yes

---

> ### Author Rebuttal · Authors · 2023-08-09
>
> We appreciate the comments and the feedback from the reviewer.
>
> W1: Technical Contribution and Novelty:
>
> To the best of our knowledge, and as stated by reviewer 1, our method is the first to use a pre-trained vision-language model for completing point clouds. That allows us to complete partial point clouds from unseen classes, and without collecting any 3D data, as required by previous approaches. Using the SDS-loss for point cloud completion is far from simple, as indicated by the ablation study in Figure 6 (main paper).
>
>
> W2: The author should not emphasize in or out-of-distribution in Figure 1:
>
> We thank the reviewer for this comment, and we will update the figure accordingly.
>
>
> W3: The authors should show results on Shapenet:
>
> Our work addresses the problem of completing point clouds captured ``in the wild”, rather than in close-world synthetic data. We expect methods trained to complete PCs of specific predefined classes on synthetic data to perform well in that setting, but as our paper demonstrates, they struggle when applied to open-world settings. We focus the experiments on real partial point cloud data captured by real sensors rather than on synthetic data sampled artificially from CAD models. Unlike the previous methods, our method was not trained on synthetic datasets so it would be unfair to base the comparisons on these datasets. Both, our method and the baselines, were not trained on real cases, so we find it fair to base the comparison on real-world data.
>
>
> Q1: Is it possible to take off the color head?
>
> We thank the reviewer for the interesting suggestion. We implemented this suggestion by setting a constant gray color to be the color of the object and changing the text prompt to contain the word “gray” in the object description. For rendering the object details we applied shading as suggested by [10]. The results are comparable to the results with the RGB branch in terms of Chamfer distance on the evaluation set (31.3mm compared to 30.5mm with the color MLP head). We conclude that such renderings are in the distribution of Stable Diffusion such that they can be used for completing the surface correctly given the signal from the point cloud. The color MLP head does help in understanding the semantic content that is being generated, especially in failure cases, but it can definitely be omitted for simplifying the network architecture if resources are limited. We will discuss this in the revised paper.

---

> > ### Comment · Reviewer_MGrG · 2023-08-10
> > **Following comment**
> >
> > Thank the authors for the reply.
> >
> > For W1, I still think the fact that this work uses a pipeline very similar to DreamFusion and Magic3D without much insight makes the novelty very limited. Also, Figure 6 shows the effectiveness of SDS loss, but since SDS loss is not a novel idea, I don't get what you mean about "far from simple". To me, I would rather call it a simple but effective trick.
> >
> > Although the novelty is very limited, I appreciate the effort in exploring new settings. Therefore, I will keep my rating for now.

---

> > > ### Author Response · Authors · 2023-08-21
> > >
> > > We thank the reviewer for the response and for the insightful discussion.
> > >
> > >
> > > ‘I don't get what you mean about "using the SDS-loss for point cloud completion is far from simple"’:
> > >
> > >
> > > The SDS loss by itself, even when combined with the SDF representation, results in inferior results. Trying to render the SDF from random views and using the SDS loss, does not complete the point cloud successfully as indicated by our qualitative and quantitative ablation study in Figure 6 (main paper),  “random cameras”. Our camera sampling is a key part that allows us to complete the object by sampling cameras in a controlled manner, starting from the viewpoint of the sensor that captured the points,  and enlarging the range gradually. This allows us to work with a non-detailed text description of the object without any diffusion model personalization techniques. By starting from the sensor’s viewpoint,  the simple description and the partial point cloud allow the completion to be compatible with both the text and the point cloud at earlier phases. Then, due to the gradual sampling, the rest of the completion consistently matches the already completed part and the text description, until the entire object is completed successfully. During this process, we verify that the diffusion model does not see an “unnatural” pose of the object, by aligning the camera coordinates with the plane that the object is located on. Technically, we verify that the horizontal axis of the camera is orthogonal to the normal of the plane. We will emphasize this contribution in Section 4.2.

---

### Official Review · Reviewer_zxYf · 2023-07-03

**Soundness:** 3 good
**Presentation:** 3 good
**Contribution:** 2 fair
**Rating:** 5
**Confidence:** 4

**Summary:**

This paper tackles point cloud completion with SDS-Complete, which aims at improving Out-Of-Distribution results with a pre-trained Stable Diffusion model for score distillation sampling (SDS). Specifically, the author propose to learn the MLPs for SDF and color with rendering-based losses with respect to the depth and the mask from the sensor's perspective. For novel views, SDS-Complete guides the rendering results with SDS which enforces a visual-language semantic. Experiments are conducted on the Redwood 3D Scans dataset with state-of-the-art results for OOD objects. The author also provides qualitative results on the KITTI LiDAR dataset.

**Strengths:**

- Figures are clear and method descriptions are detailed.
- Results of comparison experiments are encouraging.
- The motivation is clear, and the relevant pipeline design is intuitive and effective.

**Weaknesses:**

- Limited technical contribution. Including pre-trained vision-language models to improve out-of-distribution results does not bring new insights to the readers and the approach to include it (score distillation sampling in this paper) is not new.
- Limited evaluation. The experimental results are not enough to support this paper's claim. It would be good if the author can provide results on ShapeNet and other datasets.
- Weak ablation study. How much does SDS help with OOD data? It would be good if the author can provide step-by-step ablation results and more analysis on improvements with respect to OOD data.

**Questions:**

- Will the performance be affected if fewer initial points are used as the input?
- How much do the depth and mask inputs help in terms of performance?

**Limitations:**

The author commented on the limitations of SDS-Complete in terms of the huge number of sampling views and the topology of objects to be completed. It would be good if they can further specify the runtime and the number of sampling views.

---

> ### Author Rebuttal · Authors · 2023-08-09
>
> We appreciate the comments and the feedback from the reviewer.
>
> W1: Technical Contribution and Novelty:
>
> To the best of our knowledge, and as stated by reviewer 1, our method is the first to use a pre-trained vision-language model for completing point clouds. That allows us to complete partial point clouds from unseen classes, and without collecting any 3D data, as required by previous approaches. Using the SDS-loss for point cloud completion is far from simple, as indicated by the ablation study in Figure 6 (main paper).
>
>
> W2: provide results on ShapeNet
>
> Our work addresses the problem of completing point clouds captured ``in the wild”, rather than in close-world synthetic data. We expect methods trained to complete PCs of specific predefined classes on synthetic data to perform well in that setting, but as our paper demonstrates, they struggle when applied to open-world settings. We focus the experiments on real partial point cloud data captured by real sensors rather than on synthetic data sampled artificially from CAD models. Unlike the previous methods, our method was not trained on synthetic datasets so it would be unfair to base the comparisons on these datasets. Both, our method and the baselines, were not trained on real cases, so we find it fair to base the comparison on real-world data.
>
>
> W3: How much does SDS help with OOD data?
>
> Figure 6 in the main paper provides a quantitative ablation study addressing this question.  It shows that without the SDS loss, the error is significantly higher.  Following the reviewer’s request we present in a new Table 1  (Rebuttal PDF) the numbers with separated columns for in-distribution and OOD objects.
>
>
> W4: Will the performance be affected if fewer initial points are used as the input?:
>
> To address the reviewer’s question we run our method with 50% and 10% of the original input points for the evaluation set from the Redwood dataset. We found that there is no significant difference in the results (up to 1.5 mm difference in terms of average Chamfer distance). These results indicate that our method is robust to the number of input points. We will include this experiment in the revised paper.
>
>
> Q1: How much do the depth and mask inputs help in terms of performance?:
>
> Following the reviewer’s question we add to the ablation study two rows: “No Depth” and “No Mask”. The numbers are presented in Table 1 of the rebuttal PDF.  It can be seen that “No Mask” has a more significant impact on the accuracy, and that without the depth loss, the error is higher by about 10 percent.
>
>
> Q2: Further specifying the runtime and the number of sampling views:
>
> See supplementary, Section 5, “Running time”. Our test time optimization method is slow compared to the feedforward models of the baseline methods, but works much better on real point clouds, and does not require any dataset of 3D shapes for training as required by the baseline methods.

---

> > ### Comment · Reviewer_zxYf · 2023-08-19
> >
> > Thank the authors for their reply, which addressed my concern about the ablation study.
> > Albeit with limited insight, this paper made efforts toward zero-shot point completion. I do appreciate that and I think it's worth a raise in rating with more convincing experimental results.

---

> > > ### Author Response · Authors · 2023-08-21
> > >
> > > We thank the reviewer for the response and for the insightful discussion.
> > >
> > >
> > > Following a suggestion by Reviewer BnfZ, we added additional quantitative results on KITTI. We include them here as well for making it easier to follow the discussion:
> > >
> > >
> > >
> > > We follow PCN and calculated the Minimal Matching Distance (MMD). MMD is the Chamfer Distance (CD) between the output surface and the surface from ShapeNet that is closest to the input point cloud in terms of CD. We calculated this metric on the surfaces that were evaluated in our user study from two categories: car and motorcycle. These are the only categories that have associated Shapenet subsets which is a necessary condition for calculating the MMD metric. The mean MMD over the motorcycle and car shapes are presented in the table below, showing that our approach improves over baselines:
> > >
> > >
> > >
> > > |             | MMD  ↓ |
> > > |-------------|-------:|
> > > | ShapeFormer |  0.035 |
> > > | PoinTr      |  0.039 |
> > > | Ours        |  0.030 |
> > >
> > >
> > >
> > > We further computed the CLIP R-Precision metric [10] on all of our evaluated KITTI categories: “car”,”truck”,”motorcycle” and ”excavator”. This metric checks the accuracy of classifying a rendered image by choosing the class that maximizes the cosine similarity score between the image and the text: “a rendering of a <class name >” among all classes. We evaluated CLIP R-Precision on the output meshes of the different methods,  each rendered from 360 degrees with azimuth gaps of 2 degrees (180 images for each surface). We report the mean accuracy below. Here again, our approach is substantially better:
> > >
> > >
> > > |             | Accuracy (%) ↑ |
> > > |-------------|---------------:|
> > > | ShapeFormer |           50.0 |
> > > | PoinTr      |           40.6 |
> > > | Ours        |           75.7 |
> > >
> > >
> > > These metrics show that our method is better at reconstructing surfaces from partial real LiDAR scans compared with previous methods.  We will include these experiments in the revised paper.

---

### Official Review · Reviewer_Fuyp · 2023-07-04

**Soundness:** 3 good
**Presentation:** 3 good
**Contribution:** 3 good
**Rating:** 6
**Confidence:** 5

**Summary:**

This paper proposes a point cloud completion method with the help of text-to-image diffusion model and formulate the point cloud completion as a test-time optimization problem. It exploits the SDS loss proposed in Dreamfusion to generate 3D given text prompt, with a text-to-image diffusion model. Additionally, it also utilizes the supervision from the input partial point cloud (and also the depth and mask from sensor) for better regularization. And the authors also propose a camera sampling strategy to let the optimized shape follows the input point cloud. Since the model is built on top of the diffusion model's prior, it doesn't have the severe OOD problem as for previous train-test point cloud completion baselines.

**Strengths:**

By forming the task as a test-time-optimization problem, the proposed SDS-complete won't rely on training with large-scale datasets, and instead using the prior from text-to-image models,  which greatly boost the performance on OOD classes. Different from uniform camera sampling in DreamFusion which is used to generate object from scratch, the authors also proposed a camera handling procedure with strong motivation, which helps the model to fit the input partial point cloud and prevents the diffusion model to effect this geometry.

**Weaknesses:**

I think the proposed method is lack of enough ablation study. I didn't find ablation study (maybe just qualitative) of the proposed camera handling procedure. And also there might be some paper organization problem, see the questions below.

And I believe the time required is also a weakness comparing to train-test method, which I think should be discussed in paper, although currently hard to solve.

Some typos. In L231 I think it should be "multi-modal".

**Questions:**

1.I'm confusing with the epoch-iteration training schedule and the proposed camera handling. In the total 200K iterations (2k epochs x 100 iterations per epoch), are the different camera angles being sampled in a loop manner or just sample one angle for many iterations and never optimized again? I think the loop manner is the correct one, but if the rendering angles are looped again and again, I think it means the motivation of camera handling isn't that important.

2. I wonder what's the optimization time comparing with DreamFusion? Since in the point cloud completion task a stronger supervision or prior is given, I assume with shorter iterations it can get comparable geometry? (Although definitely more iterations the results can be better).

3. Some results seems to be unexpected. Like the second row in Figure3. The chair leg with point cloud is intermittent, but the leg behind without point cloud supervision is perfect. Is it because the point cloud is sparse and sometimes the L_p can have bad influence?

4. Is the low resolution 80x80 a bottleneck? I remeber it isn't the render resolution in DreamFusion.

**Limitations:**

Yes, the authors have a discussion of their limitations.

---

> ### Author Rebuttal · Authors · 2023-08-09
>
> We appreciate the comments and the feedback from the reviewer.
>
> W1:  Missing Ablation study for camera handling:
>
> Figure 6  in the main paper shows a table with a quantitative ablation study on the camera handling procedure (“Random camera: running without our camera handling that is described in Sec. 4.2”). It shows that without our camera sampling, the mean Chamfer error is significantly larger.
>
>
> W2:  Time required is also a weakness that should be discussed:
>
> See the discussion of the limitations in L258. Our test time optimization method is slow compared to the feedforward models of the baseline methods but its quality is far superior on real point clouds, and does not require any dataset of 3D shapes for training as required by the baseline methods.  We will emphasize this point more clearly in the revised version
>
>
> Some typos In L231
>
> We will fix this in the revised paper. Thank you.
>
>
> Q1: Camera Sampling Missing Details:
>
> L76 in the SM gives more implementation details. At each iteration, the cameras are sampled randomly from a uniform distribution with a limited range. This range increases during training, where the full range is applied from epoch 120. That allows the color and the geometry to be optimized consistently and gradually, starting from the areas that are covered by the sensor, to the areas without any input points. See the ablation study in Figure 6 of the main paper for quantitative and qualitative ablation studies on camera sampling.
>
>
> Q2 What is the optimization time compared to DreamFusion?
>
> Most of the time is consumed by pushing gradients for the SDS loss.
> By default, DreamFusion is trained for 100 epochs. In our submitted version we let our method run for 2000 epochs. Following the reviewer’s question, we present in Table 2 (Rebuttal PDF) the effect of running our method for a shorter time. It can be seen that after 5% (100 epochs) of the total running time,  our method already outperforms the baseline in terms of average Chamfer distance.  Given more running time reduces our error rate even more.
>
>
> Q3: Intermittent patterns in the legs of the chair
>
> The thin structure of the input points is too weak for constraining the surface of the outside chair presented in Figure 3.  Unfortunately, the SDS loss does not help in making the surface thicker since the rendering of the thin surface looks valid. This is due to our rendering process that uses VOLSDF which defines a smooth mapping from surface to density (Equation 5), and since the entire leg has SDF values that are close to 0, the legs get densities that produce valid low-resolution renderings. We demonstrate the issue in Figure 1 (Rebuttal PDF).   We will discuss this limitation in the revised version of the paper.
>
>
> Q4: Is the low rendering resolution 80x80 a bottleneck?
>
> DreamFusion is trained with 64x64 rendered images. In general, higher resolution requires higher GPU memory where 64x64,80x80,128x128 require 16GB(Google Colab), 32GB(V100), and 80GB(A100)  memory respectively.

---

> > ### Comment · Reviewer_Fuyp · 2023-08-12
> >
> > Thanks the authors for detailed reply. And I have also checked other reviewers' opinions and the authors' response.
> >
> > I think the rebuttal answers my concerns. And I would like to see the epoch ablation part and the discussion of rebuttal's fig.1 be shown in the revised version. The VolSDF formulation entangled with the low-resolution rendering in this case reveals some underlying problem and I would like to see them be discussed in paper.
> >
> > I also agree with other reviewers' opinions that the idea is with limited novelty. But I also think this work is with high degree of completion, and also shows some new scenarios in the certain case. For that, I would keep my rating as weak accept but a little prone to borderline side, any discussion is welcomed.

---

> > > ### Author Response · Authors · 2023-08-21
> > >
> > > We thank the reviewer for the response and for the insightful discussion.
> > >
> > >
> > > Epoch ablation and rebuttal's fig.1
> > >
> > >
> > > We will add these results to the revised version with a proper discussion.
> > >
> > >
> > > “Limited novelty”
> > >
> > >
> > > To the best of our knowledge, our method is the first to use a pre-trained vision-language model for completing point clouds. That allows us to complete partial point clouds from unseen classes, and without collecting any 3D data, as required by previous approaches. The SDS loss by itself, even when combined with the SDF representation, results in inferior results. Trying to render the SDF from random views and using the SDS loss, does not complete the point cloud successfully as indicated by our qualitative and quantitative ablation study in Figure 6 (main paper),  “random cameras”. Our camera sampling is a key part that allows us to complete the object by sampling cameras in a controlled manner, starting from the viewpoint of the sensor that captured the points,  and enlarging the range gradually. This allows us to work with a non-detailed text description of the object without any diffusion model personalization techniques. By starting from the sensor’s viewpoint,  the simple description and the partial point cloud allow the completion to be compatible with both the text and the point cloud at earlier phases. Then, due to the gradual sampling, the rest of the completion consistently matches the already completed part and the text description, until the entire object is completed successfully. During this process, we verify that the diffusion model does not see an “unnatural” pose of the object, by aligning the camera coordinates with the plane that the object is located on. Technically, we verify that the horizontal axis of the camera is orthogonal to the normal of the plane. We will emphasize this contribution in Section 4.2.

---

### Official Review · Reviewer_BnfZ · 2023-07-06

**Soundness:** 1 poor
**Presentation:** 3 good
**Contribution:** 2 fair
**Rating:** 4
**Confidence:** 5

**Summary:**

This paper proposed a novel method for completing a 3D object from its incomplete input shape. It specifically focuses on the out-of-distribution objects and proposed a diffusion model based framework to generatively learns a SDF for the complete shapes. The idea is simple and easy to follow.

**Strengths:**

The idea of using cross-modal generative network to complete partial shape is novel.
The writing and organization is good.

**Weaknesses:**

1. The completion of out-of-distribution objects aims for practical usage. However, the proposed SDS-Complete requires up to 5 different inputs for completing a partial shape, which is contrary to the original intention for out-of-distribution shape completion. Specifically, according to the description in Sec.4, SDS-Complete requires: 1) depth image; 2) segmented point cloud; 3) segmented binary mask; 4) representation of text; 5) internal parameters of the sensor, and each of them is indispensable. Therefore, the proposed method is highly impractical for real-world scenario compared with image reconstruction or shape completion from single input.

2. As for in-domain completion results visualized in Figure 3, the complete shape generated by PoinTr is highly suspicious and may be unfair. The chair is one of the most commonly used categories in shape completion experiments, and PoinTr has been referred by many previous studies to generate a robust prediction on this chair category. However, in Figure 3, PoinTr even fails to predict the missing chair legs, which in reviewer's opinion, is highly impossible if the training procedure is correct.

3. According to the above discussion of visualization results, the quantitative results in Table 1 are insufficient to prove the effectiveness of the proposed method in terms of out-of-domain shape completion task.

4. Moreover, in-domain comparison should at least be conducted on one of the popular completion benchmarks such as ShapeNet dataset, MVP dataset or PCN dataset, but none of them appear in the experiments.

5. The quantitative results on KITTI dataset is missing, only 5 samples in Figure 5 are highly insufficient to verify the effectiveness of the proposed method on KITTI dataset.


**Questions:**

See weaknesses

**Limitations:**

The limitations has been addresses in the draft.

---

> ### Author Rebuttal · Authors · 2023-08-09
>
> We appreciate the comments and the feedback from the reviewer.
>
>
> W1: The method requires 5 input types
>
> Our setup follows an existing protocol that is presented in ShapeFormer which uses real partial point clouds that are extracted from real-world sensors. In contrast to synthetic setups where each point cloud is artificially sampled from a single CAD model, the evaluated real point clouds are recorded by an active depth sensor (LiDAR or depth camera) that stores for each ray the distance of the point from the sensor location.  Therefore, depth information is not a limitation for real point clouds since it is inherently available in real PC data and does not limit the method in practice. The internal parameters for each sensor are available online.
> Note that these real sensors record a complete scene and therefore for isolating the relevant object from the scene (for any of the baseline methods), it is required to segment out the object from the scene.  Regarding text input, some other completion methods [3-6] use different trained models for each class (tables, chairs, …), thus they assume that the object class is known.
>
> W2: Complete shape generated by PoinTr is highly suspicious and may be unfair:
>
> Note that PoinTr was trained and evaluated on synthetic data. Therefore its generalization to real data is limited. Moreover, PoinTr has some generalization issues even for Shapenet models e.g. Figure 4 in [7].
>
>  W3: The quantitative results in Table 1 are insufficient to prove the effectiveness of the proposed method in terms of out-of-domain shape completion task
>
> According to Table 1, we improve over the baseline methods by 50% on real-world point clouds. Note that additional qualitative and quantitative examples are presented in the supplementary materials.
>
> W4: Missing comparisons on Shapenet/MVP/PCN
>
> Our work addresses the problem of completing point clouds captured ``in the wild”, rather than in close-world synthetic data. We expect methods trained to complete PCs of specific predefined classes on synthetic data to perform well in that setting, but as our paper demonstrates, they struggle when applied to open-world settings. We focus the experiments on real partial point cloud data captured by real sensors rather than on synthetic data sampled artificially from CAD models. Unlike the previous methods, our method was not trained on synthetic datasets so it would be unfair to base the comparisons on these datasets. Both, our method and the baselines, were not trained on real cases, so we find it fair to base the comparison on real-world data.
>
> W5 : Comparisons on KITTI are quantitative using only 5 samples
>
> We conducted a user study on 15 sampled KITTI objects. The results are presented in the supplemental material, showing that our method outperformed the baseline methods in 73.4% of the cases. These results demonstrate that our method outputs better completions on KiTTI in terms of quality. Since GT shapes for KiTTI are not available, it is not possible to compute automated metrics.

---

> > ### Comment · Reviewer_BnfZ · 2023-08-18
> >
> > Thanks for the detailed reply.
> > The reviewer is satisfied with some of the replies, but still got unsolved concerns. I would like to raise my ratings but still lean to rejection.
> > The number of KITTI samples for qualitative comparison is not the key problem, but the quantitative comparison. Many previous methods like PCN or Pcl2Pcl have introduced some metrics to evaluate results without GT for dataset like KITTI.  On the other hand, the response in W1 still do not fully address the problem of requiring so much input, some of which are not always accessible in practice and will limit the implementation of the proposed method. For example, the internal parameters may not be easily found on some real scan dataset like ScanNet v2 or S3DIS, not to mention in industrial scenarios.

---

> > > ### Author Response · Authors · 2023-08-21
> > >
> > > We thank the reviewer for the response and for the insightful discussion.
> > >
> > >
> > > The key remaining concern is that quantitative results for KITTI are missing.
> > >
> > >
> > >
> > > Thank you for your insight and for providing concrete suggestions for evaluation metrics that allow us to evaluate KITTI even without GT data. Following the reviewer's suggestions, we follow PCN and calculated the Minimal Matching Distance (MMD). MMD is the Chamfer Distance (CD) between the output surface and the surface from ShapeNet that is closest to the input point cloud in terms of CD. We calculated this metric on the surfaces that were evaluated in our user study from two categories: car and motorcycle. These are the only categories that have associated Shapenet subsets which is a necessary condition for calculating the MMD metric. The mean MMD over the motorcycle and car shapes are presented in the table below, showing that our approach improves over baselines:
> > >
> > >
> > >
> > > |             | MMD  ↓ |
> > > |-------------|-------:|
> > > | ShapeFormer |  0.035 |
> > > | PoinTr      |  0.039 |
> > > | Ours        |  0.030 |
> > >
> > >
> > >
> > > We further computed the CLIP R-Precision metric [10] on all of our evaluated KITTI categories: “car”,”truck”,”motorcycle” and ”excavator”. This metric checks the accuracy of classifying a rendered image by choosing the class that maximizes the cosine similarity score between the image and the text: “a rendering of a <class name >” among all classes. We evaluated CLIP R-Precision on the output meshes of the different methods,  each rendered from 360 degrees with azimuth gaps of 2 degrees (180 images for each surface). We report the mean accuracy below. Here again, our approach is substantially better:
> > >
> > >
> > > |             | Accuracy (%) ↑ |
> > > |-------------|---------------:|
> > > | ShapeFormer |           50.0 |
> > > | PoinTr      |           40.6 |
> > > | Ours        |           75.7 |
> > >
> > >
> > > These metrics show that our method is better at reconstructing surfaces from partial real LiDAR scans compared with previous methods.  We will include these experiments in the revised paper.
> > >
> > >
> > >  “The internal parameters may not be easily found on some real scan datasets”
> > >
> > >
> > > We agree that requiring the internal parameters is a limitation of our method. We will discuss this limitation in the paper. With that said, it is important to note that any method for processing point clouds needs the camera's internal parameters for extracting a point cloud from a depth image.

---

### Official Review · Reviewer_adEs · 2023-07-06

**Soundness:** 3 good
**Presentation:** 3 good
**Contribution:** 3 good
**Rating:** 6
**Confidence:** 2

**Summary:**

This work proposes to use text-to-image diffusion model for OOD point-cloud completion. Similar to DreamFusion3D that trains NeRF accompanied by an SDS loss for 3D generation, this work applies the idea to point-cloud completion. Experiments show that the performance is good on both Redwood dataset and KITTI dataset.

**Strengths:**

I think the proposed idea is simple yet novel. The results also show the effectiveness of the proposed method. The paper is also easy to follow.

**Weaknesses:**

I notice in both figure 3 and figure 5, although the patterns/shape of the completed point-cloud is correct, it seems that they are not smooth which even makes it worse than ShapeFormer in Figure 3. This makes the performance of the proposed idea less attractive, despite the paper targets at OOD point-cloud completion.

**Questions:**

If I understand correctly, the method requires to overfit per-object via NeRF, on the other hand, since it requires inference pre-trained SD for each iteration, it seems that the training process is pretty slow. Can the authors provide the training details?

**Limitations:**

It seems that this work can only work for the cases of single objects since it heavily relies on the capability of pre-trained stable diffusion. I wonder whether it can achieve scene-level completion.

---

> ### Author Rebuttal · Authors · 2023-08-09
>
> We appreciate the comments and the feedback from the reviewer.
>
> W1: Generated completions are not smooth in Figure 3 and Figure 5
>
> Overall our method produces better outputs qualitatively (see more examples in the SM, Figure 4,5 ),  and quantitatively where our method’s Chamfer distance is lower by 50% overall, and by 30% for Shapenet classes (see Table 1 )  compared to ShapeFormer. Regarding Figure 5, in the user study (SM, Section 3),  for 73.4% of the cases, the participants preferred our outputs over ShapeFormer’s outputs, which demonstrates that our method outputs better completions in terms of quality.
>
>
> Q1: Provide details on running time:
>
> See supplementary, Section 5, “Running time”. Our test time optimization method is slow compared to the feedforward baseline methods,  but works much better on real point clouds, and does not require any dataset of 3D shapes for training as required by the baseline methods. We believe that it could be accelerated  e.g. by combining recent acceleration techniques such as Instant-NGP but this is left as a future work.
>
>
> L1: The method is limited to single-object:
>
> The reviewer is correct. There are natural ways to extend this approach to scenes with multiple objects. For instance, starting with segmenting a scene, and splitting it into a disjoint set of partial point clouds. This tends to be easier if RGB appearance data is available. Then, our method can reconstruct each object separately, and the results can be merged back together to create the scene. Interestingly, our loss can be extended to take into account segmentation confidence. This is left for future work.

---

> > ### Comment · Reviewer_adEs · 2023-08-14
> >
> > Thanks for the reply. The authors address my concerns. But I still think the test-time overfitting is too slow. I think the overall work is interesting and I will keep my rate.

---

> > > ### Author Response · Authors · 2023-08-21
> > >
> > > We thank the reviewer for the response and for the insightful discussion. The slow running time is indeed a limitation, and we will extend the discussion about this point in the revised paper. Furthermore, we will add Table 2 from the Rebuttal PDF to the revised version that demonstrates the effect of running our method for shorter times and specifically shows that after 5% (100 epochs) of the total running time,  our method already outperforms the baseline in terms of average Chamfer distance.
> > > We do note that our method can solve in-the-wild cases that cannot be solved by existing methods without gathering additional data. Therefore, we believe that acceleration can remain a future challenge.

---

### Author Rebuttal · Authors · 2023-08-09

We thank the reviewers for their insightful comments. Below we answer separately for each reviewer.

---

### Decision · Program_Chairs · 2023-09-21

**Decision:**

Accept (poster)

**Comment:**

The scores are mix. The AC carefully read all the review comments and the authors' responses, and think evenly the setting requires more inputs and heavy computing cost. While using 2D prior model for point completion task is very new, and the results are also promising. Thus, The AC vote to accept.